# Bayesian Model Selection Approach to Boundary Detection with Non-Local Priors

**Fei Jiang**
Department of Statistics and Actuarial Science
The University of Hong Kong
`feijiang@hku.hk`

**Guosheng Yin**
Department of Statistics and Actuarial Science
The University of Hong Kong
`gyin@hku.hk`

**Dominici Francesca**
Harvard T.H. Chan School of Public Health
Harvard University
`fdominic@hsph.harvard.edu`

## Abstract

Based on non-local prior distributions, we propose a Bayesian model selection (BMS) procedure for boundary detection in a sequence of data with multiple systematic mean changes. The BMS method can effectively suppress the non-boundary spike points with large instantaneous changes. We speed up the algorithm by reducing the multiple change points to a series of single change point detection problems. We establish the consistency of the estimated number and locations of the change points under various prior distributions. Extensive simulation studies are conducted to compare the BMS with existing methods, and our approach is illustrated with application to the magnetic resonance imaging guided radiation therapy data.

## 1 Introduction

Traditional change point detection algorithms often apply to the situation where the occurrence frequency of the change points is relatively consistent across the signals. For example, the narrowest-over-threshold (NOT) algorithm [1] is more suitable when different segments between the change points have comparable lengths, and the stepwise marginal likelihood (SML) method [5] works better to identify frequent change points. However, in practice it is often the case that distances between consecutive change points may vary dramatically, while only those with certain distance gaps are of interest. For such settings, we develop a computationally efficient Bayesian model selection (BMS) approach to identifying multiple change points.

The inconsistent gaps between the change points can be observed from the signals generated by the magnetic resonance imaging guided radiation therapy (MRgRT). When radiations travel in the magnetic field, the dose can be significantly enhanced near the boundaries between different tissues or organs inside human bodies. As shown by Figure 1, the Duke Mid-sized Optical-CT System (DMOS) is developed to identify the dose changes near the region of such boundary artifact. It also exhibits the profile of dose intensities as the radiation travels through the dosimeter, where the boundaries on and inside the dosimeter can be distinguished by the notable peaks in the signals. In the experiment, radiations enter the cylindrical dosimeter from different directions, and a sequence of dose intensities ordered by their distances to the sources are recorded. Because the dosimeter is circular and there is a

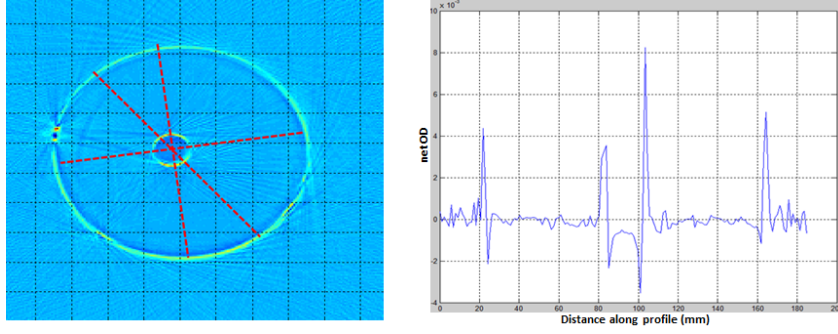

Figure 1: Reconstructed image of a slice in a cylindrical dosimeter with a cavity in the middle (left) and a typical line profile through the center of the cavity (right). Radiations enter the dosimeter from the hole on the left of the cylindrical dosimeter, which rotates 360 degrees so that radiations can enter from different directions.

cavity in the middle, radiations from different directions would hit the boundaries in the dosimeter at similar distances from their sources.

In the MRgRT data, radiations in certain directions may experience temporary changes at non-boundary locations, which may result from the abnormal status of the DMOS system rather than the true dose changes. The temporary change points, appearing in the data sequence as the spike points, are often mixed up with those on the boundary (i.e., the peak locations in the right panel of Figure 1), which makes the boundary detection extremely challenging. Figures A.1 in Appendix shows the change points in the MRgRT data identified by the NOT [1] and SML [5] algorithms respectively, while neither can correctly identify the true boundaries. This motivates us to propose a new approach to detecting the systematic changes when the segment lengths have dramatic differences.

Our preliminary analysis of the MRgRT data demonstrates that the local control of the discovery is crucial. To avoid picking the spike points, we enforce a minimal distance between adjacent change points. Moreover, we adopt a computationally efficient local scan routine and propose a systematic two-stage procedure to speed up the change point detection. More specifically, the local scan method first identifies the candidate points with a minimal distance based on the local data, and then optimizes an utility function to obtain the estimates for the locations and the total number of change points. Because the change points are defined based on the mean changes between two consecutive segments, the local data are sufficient to detect the systemic changes [6, 7, 8, 13, 14, 16, 18, 19].

To perserve the positive detection rate of the change points and reduce the false detection rate of the non-change points, we take a Bayesian marginal likelihood function as the utility, and develop a new BMS procedure for identifying change points. We show that the selection consistency is achieved under both the local [2, 3, 4, 17] and non-local priors [11], whereas the convergence rate is faster under the later. Our BMS procedure is cast in the model selection framework, which is faster than the dynamic programming in the SML framework. For example, for the MRgRT data, BMS takes 1.3 seconds and SML takes 3.1 seconds when the maximum number of change points is capped at 100. The efficiency of BMS is mainly due to the fact that it reduces the search space dramatically by selecting a small set of candidate change points. Once the candidate points are selected, BMS only needs to evaluate two consecutive segments at a time, which greatly facilitates parallel computation.

## 2 Bayesian multiple change points detection

### 2.1 Probability model

Suppose there are $p_0$ true change points $t_1 < \cdots < t_{p_0}$ among $n$ observations $\mathbf{Y}_n = \{Y_1, \ldots, Y_n\}$. As a convention, let $t_0 = 1$ and $t_{(p_0+1)} = n+1$. Denote $\lambda_j = t_{j+1} - t_j$ and $\lambda = \min_{j=0,\ldots,p_0} \lambda_j$. We consider a set of $K_n$ candidate points $\tau_1, \ldots, \tau_{K_n}$, with $\tau_0 = 1$ and $\tau_{K_n+1} = n+1$, while selection of the candidate points is discussed in Section 2.3. Define $n_j = \tau_{j+1} - \tau_j$, and $n_I = \min_{j=0,\ldots,K_n-1} n_j$, $n_I \leqslant \lambda$. Let $\mathcal{H}(n_I) = \{\tau_j : j = 1, \ldots, K_n, |\tau_{j+1} - \tau_j| > n_I\}$ denote the set of candidate change points, and let $\mathcal{T}_0(p_0) = \{t_j : j = 1, \ldots, p_0\}$ denote the set of true change points. Not only does the specification of the candidate points allow BMS to be implemented in a lower dimensional space with the most influential points, but it also guarantees that there are a sufficient number of non-change points surrounding the true change points so that the consistency conditions are met. The probability

model takes the form of

$$Y_l = \nu_{\tau_j} + \epsilon_l, \quad l \in [\tau_j, \tau_{j+1}),$$

where the random errors $\epsilon_l$ are independent with mean zero and variance $\sigma_j^2$. Further, we define $\sigma = \max_{j=0,\ldots,p_0} \sigma_j$.

For ease of exposition, we first consider the case where the locations of the candidate change points are given and $\mathcal{T}_0(p_0) \subset \mathcal{H}(n_I)$. Define $\bar{Y}_{\tau_j} = n_{j-1}^{-1} \sum_{l=\tau_{j-1}}^{\tau_j-1} Y_l$, which is the sample average for the $(j-1)$th segment $[\tau_{j-1}, \tau_j)$, $j > 1$. If the candidate point $\tau_k$ is not a change point, then the points in $[\tau_k, \tau_{k+1})$ should have the same mean as those in $[\tau_{k-1}, \tau_k)$; otherwise there should be a mean shift between the segments $[\tau_k, \tau_{k+1})$ and $[\tau_{k-1}, \tau_k)$. Hence, we can formulate the model and prior distribution for $l \geqslant \tau_1$ as follows:

$$
\begin{aligned}
Y_l - \bar{Y}_{\tau_k} &= \mu_k + \xi_l, \quad l \in [\tau_k, \tau_{k+1}), \\
\mu_k &\sim \pi(\mu_k), \quad \text{if } \tau_k \text{ is a change point,} \\
\mu_k &= 0, \quad \text{with probability 1, if } \tau_k \text{ is an } n_I\text{-flat point,}
\end{aligned}
$$

where $\xi_l$ is a mean-zero error term and $\pi(\cdot)$ is a prior distribution. The $n_I$-flat point is defined as a non-change point which is at least $n_I$ apart from any change points. We require the $n_I$ distance between the true change points and the flat ones so that there are sufficient neighborhood samples to achieve the estimation consistency.

Let $\mu_{k0}$ be the true value of $\mu_k$, and we assume $|\mu_{k0}| > \delta$, where $\delta > 0$ is the lower bound of $\mu_{k0}$, for the $k$'s with $\tau_k \in \mathcal{T}_0(p_0)$. The prior distribution of $\mu_k$ determines the convergence rate of the BMS procedure. We explore three types of priors: the local prior [9], the non-local moment prior and the inverse moment prior [11] as follows:

$$
\begin{aligned}
\text{Local prior: } \pi_L(\mu) &= N(0, \omega^2), \\
\text{Moment prior: } \pi_M(\mu) &= \mu^{2v}/C_M 1/\sqrt{2\pi} \exp(-\mu^2/2), \\
\text{Inverse moment prior: } \pi_I(\mu) &= s\nu^{q/2}/\Gamma(q/2s)\mu^{-(q+1)} \exp\left\{-(\mu^2/\nu)^{-s}\right\},
\end{aligned}
$$

where $C_M$ is the normalizing constant.

Let $M_k$ represent the model that $\tau_k$ is the sole change point. We define the marginal likelihood with the Gaussian kernel as

$$\Pr(\mathbf{Y}_n|M_k) = \prod_{j=1,j\neq k}^{K_n} \prod_{l=\tau_j}^{\tau_{j+1}-1} \exp\{-(Y_l - \bar{Y}_{\tau_j})^2\} \int \prod_{l=\tau_k}^{\tau_{k+1}-1} \exp\{-(Y_l - \bar{Y}_{\tau_k} - \mu)^2\}\pi(\mu)d\mu.$$

The posterior model probability of $M_k$ given $\mathbf{Y}_n$ is

$$\Pr(M_k|\mathbf{Y}_n) = \frac{\Pr(\mathbf{Y}_n|M_k)\Pr(M_k)}{\sum_{j=1}^{K_n}\Pr(\mathbf{Y}_n|M_j)\Pr(M_j)} = \frac{\Pr(\mathbf{Y}_n|M_k)}{\sum_{j=1}^{K_n}\Pr(\mathbf{Y}_n|M_j)},$$

when $M_j$ takes a discrete uniform prior, $j = 1, \ldots, K_n$. It is not necessary for $\mathbf{Y}_n$ to be normally distributed to ensure the selection consistency in detecting mean changes, while the Gaussian kernel is used here because it tends to be large when the difference between the true and the hypothetical segment means is small. Hence, as $n \to \infty$, $\Pr(M_k|\mathbf{Y}_n)$ approaches 1 when $\tau_k$ is a true change point and the $\tau_j$'s $(j \neq k)$ are $n_I$-flat points.

## 2.2 Detection of change points

We start with the simplest case where there is only one mean shift in the data, i.e., $p_0 = 1$ is fixed *a priori*. We select the candidate point $\tau_k$ corresponding to the largest $\Pr(M_k|\mathbf{Y}_n)$, i.e., the largest marginal likelihood $\Pr(\mathbf{Y}_n|M_k)$. It can be shown that

$$\Pr(M_k|\mathbf{Y}_n) = \left\{1 + \sum_{\substack{j=1\\j\neq k}}^{K_n} \frac{\Pr(\mathbf{Y}_n|M_j)}{\Pr(\mathbf{Y}_n|M_k)}\right\}^{-1},$$

where for $j \neq k$,

$$\Pr(\mathbf{Y}_n|M_j) = \frac{\int \prod_{l=\tau_j}^{\tau_{j+1}-1} \exp\{-(Y_l - \bar{Y}_{\tau_j} - \mu)^2\}\pi(\mu)d\mu}{\prod_{l=\tau_j}^{\tau_{j+1}-1} \exp\{-(Y_l - \bar{Y}_{\tau_j})^2\}},$$

and for $j = k$ we replace above $\tau_j$ and $\tau_{j+1}$ by $\tau_k$ and $\tau_{k+1}$ respectively. As a result, the selection consistency is determined by the evidence in favor of $\mu_k \sim \pi(\mu_k)$ and $\mu_j = 0$ for $j \neq k$.

For the case with multiple change points ($p_0 > 1$), we select the points corresponding to the $p_0$ largest $\Pr(M_k|\mathbf{Y}_n)$, for which the selection consistency is presented as follows.

**Theorem 1.** *Let* $\mathcal{M} = \{M_k, \tau_k \in \mathcal{T}_0(p_0)\}$. *If it holds that*

$$\Pr(\mathbf{Y}_n|M_j) = O_p(a_{n_j}), \tag{1}$$

*for* $\tau_j \notin \mathcal{T}_0(p_0)$, $a_{n_j} = o_p(1)$, *and* $n_I^{1/2}\delta/\sigma \to \infty$, *then*

$$\sum_{M_k \in \mathcal{M}} \Pr(M_k|\mathbf{Y}_n) = 1 + O_p\{K_n a_{n_I} \exp(-n_I \delta^2)\}.$$

*Hence, as* $n_I/\log(n) \to c > 0$, $n_I \leqslant \lambda$, *we have*

$$\sum_{M_k \in \mathcal{M}} \Pr(M_k|\mathbf{Y}_n) \xrightarrow{p} 1.$$

The proof of Theorem 1 is delineated in Appendix. The selection consistency depends on the convergence rate of $a_{n_I}$, which is determined by the prior $\pi(\cdot)$. Lemmas 2–4 in Appendix show that $a_{n_j} = n_j^{-1/2}$ for local prior $\pi_L(\mu)$; $a_{n_j} = n_j^{-v-1/2}$ for $\pi_M(\mu)$ and $a_{n_j} = \exp\{-n_j^{s/(s+1)}\}$ for $\pi_I(\mu)$. Hence, the selection consistency is achieved at the fastest rate using the non-local inverse moment prior.

When $p_0$ is unknown, let $\mathcal{T}(p)$ be the set containing $p$ points obtained by the procedure described above. We define the marginal likelihood given $\mathcal{T}(p)$ as

$$\Pr\{\mathbf{Y}_n|\mathcal{T}(p)\} = \prod_{\tau_j \notin \mathcal{T}(p)} \prod_{l=\tau_j}^{\tau_{j+1}-1} \exp\{-(Y_l - \bar{Y}_{\tau_j})^2\}$$

$$\times \prod_{\tau_k \in \mathcal{T}(p)} \int \prod_{l=\tau_k}^{\tau_{k+1}-1} \exp\{-(Y_l - \bar{Y}_{\tau_k} - \mu)^2\}\pi(\mu)d\mu.$$

We can estimate the locations and the number of change points in two steps: First for any given $p$, we obtain $\widehat{\mathcal{T}}(p)$ using the procedure described in the previous section; and second we estimate $p_0$ by $\widehat{p}$ by maximizing $\Pr\{\mathbf{Y}_n|\widehat{\mathcal{T}}(p)\}$ with respect to $p$, which is merely implemented in one dimension. In contrast, SML [5] simultaneously estimates the locations and the number of change points by maximizing the marginal likelihood with respect to both $\mathcal{T}(p)$ and $p$.

## 2.3 Selection of candidate points

Previous discussions rely upon a critical assumption that the candidate points are specified in advance. To facilitate the implementation of BMS, we need to find a candidate set $\mathcal{H}_c(n_I)$ that is close to $\mathcal{H}(n_I)$. For the selection consistency of the change points, we require for each $t_j$ there is a $\tau_k \in \mathcal{H}_c(n_I)$, such that $\Pr(|t_j - \tau_k| \leqslant n_I) = 1 - O_p[\min\{\exp(-n_I\delta^2), a_{n_I}\}]$. Define

$$R_i = \frac{\int \prod_{l=i}^{i+n_I-1} \exp\{-(Y_l - \bar{Y}_i - \mu)^2\}\pi(\mu)d\mu}{\prod_{l=i}^{i+n_I-1} \exp\{-(Y_l - \bar{Y}_i)^2\}},$$

where $\bar{Y}_i = n_I^{-1}\sum_{j=i-n_I}^{i-1} Y_j$. By the argument similar to that in Lemma 1, $R_i$ goes to infinity when $i$ is a true change point, and $R_i$ approaches zero in probability when $i$ is an $n_I$-flat point. Hence, the value of $R_i$ can distinguish a change point from a set of $n_I$-flat points. To further eliminate the non-change points that are also not $n_I$-flat, we implement the non-maximum suppression that removes the points which do not yield the largest $R_i$'s in their $n_I$-neighborhood. Specifically, the screening procedure for selecting candidate points is described as follows.

---
**Algorithm 1** : **Screening**

---

    (i) For each $i$ in $[n_I, n - n_I]$, compute $R_i$.

   (ii) If $R_i = \max\{R_j : j \in (i - n_I, i + n_I]\}$, then $i$ is selected as a candidate point.

  (iii) Scan through the entire data sequence, and obtain a set of $K_n$ candidate points $\mathcal{H}_c(n_I)$.

---

The screening algorithm is comparable to that in [15], as by the Laplace approximation we have

$$
\begin{aligned}
R_i &= \frac{D_n \prod_{l=i}^{i+n_I-1} \exp\{-(Y_l - \bar{Y}_i - \mu)^2\}\pi(\check{\mu})}{\prod_{l=i}^{i+n_I-1}\exp\{-(Y_l - \bar{Y}_i)^2\}}\{1 + o_p(1)\} \\
&= D_n \exp\left\{2\left(\sum_{l=i}^{i+n_I-1} Y_l - \sum_{j=i-n_I}^{i-1} Y_j\right)\check{\mu} - n_I\check{\mu}^2\right\}\pi(\check{\mu})\{1 + o_p(1)\},
\end{aligned}
$$

where $D_n$ is a constant of order $O_p(n_I^{-1/2})$ and $\check{\mu}$ is the maximizer of $-\sum_{l=i}^{i+n_I-1}(Y_l - \bar{Y}_i - \mu)^2 + \log\pi(\mu)$. The magnitude of the leading term in $R_i$ is strongly associated with $n_I^{-1}(\sum_{l=i}^{i+n_I-1} Y_l - \sum_{j=i-n_I}^{i-1} Y_j)$, which is the local diagnosis function with $h = n_I$ in [15].

The screening procedure identifies a candidate set $\mathcal{H}_c(n_I)$ that would lead to the consistency result.

**Proposition 1.** *Assume that $n_I^{1/2}\delta/\sigma \to \infty$, and for each $t_j \in \mathcal{T}_0(p_0)$, there is a $\tau \in \mathcal{H}_c(n_I)$, such that $\Pr\{t_j \in (\tau - n_I, \tau + n_I)\} = 1 - O[\min\{\exp(-n_I\delta^2), a_{n_I}\}]$.*

In theory, $i = t_j$ maximizes $R_i$ in the $n_I$-neighborhood of $t_j$ asymptotically. By selecting the local maximal $R_i$ in the screening procedure, $\mathcal{H}_c(n_I)$ would cover the $n_I$-neighborhood of $\mathcal{T}_0(p_0)$ as $n \to \infty$. Also the condition $n_I^{1/2}\delta/\sigma \to \infty$ indicates that the effect size cannot be too small in order to find the candidate points around the true change points. After selecting the candidate points, we perform a refinement step to identify the locations and the total number of change points.

---

**Algorithm 2** : **Refinement**

**Scanning**

    (i)  Compute $\Pr(\mathbf{Y}_n|M_k)$ by scanning over all the candidate points in $\mathcal{H}_c(n_I)$.

   (ii)  For each $p$, obtain a set of change points $\widehat{\mathcal{T}}(p)$ corresponding to the $p$ largest $\Pr(\mathbf{Y}_n|M_k)$, $k = 1, \dots, K_n$.

**Optimization**

  (iii)  Select $\widehat{p}$ that maximizes $\Pr\{\mathbf{Y}_n|\widehat{\mathcal{T}}(p)\}$.

---

**Theorem 2.** *Assume that $n_I/\log(n) \to c > 0$, $n_I^{1/2}\delta/\sigma \to \infty$, $\limsup_{n\to\infty} n_I/\lambda < 1/2$, and (1) holds. Let $\mathcal{H}_c(n_I)$ be the set of candidate points such that $|\tau_{k+1} - \tau_k| > n_I$, and for each $t_j$ there is a $\tau_k \in \mathcal{H}_c(n_I)$, $\Pr(|t_j - \tau_k| \leqslant n_I) = 1 - O_p[\min\{\exp(-n_I\delta^2), a_{n_I}\}]$. Then,*

$$
\Pr(\widehat{p} = p_0) = 1 - O_p[\max\{\exp(-n_I\delta^2), a_{n_I}\}],
$$

*and furthermore,*

$$
\Pr\left\{\sup_{\widehat{t}_j \in \widehat{\mathcal{T}}(\widehat{p})} \inf_{t_j \in \mathcal{T}_0(p_0)} |(\widehat{t}_j - t_j)/n| \leqslant n_I/n\right\} = 1 - O\{\exp(-n_I\delta^2)\},
$$

$$
\Pr\left\{\sup_{t_j \in \mathcal{T}_0(p_0)} \inf_{\widehat{t}_j \in \widehat{\mathcal{T}}(\widehat{p})} |(\widehat{t}_j - t_j)/n| < n_I/n\right\} = 1 - O(a_{n_I}).
$$

Theorem 2 shows that BMS controls both the over- and under-segmentation errors. The rationale is that for any $\mathcal{T}(p)$ different from $\mathcal{T}_0(p_0)$, there is at least a chosen point $\tau \in \mathcal{T}(p)$ whose $n_I$-neighborhood does not contain true change points. Then the likelihood ratio $\Pr\{\mathbf{Y}_n|\mathcal{T}(p)\}/\Pr\{\mathbf{Y}_n|\mathcal{T}_0(p_0)\}$ goes to 0 with probability 1, because the ratio contains at least one of $\Pr(\mathbf{Y}_n|M_j)$ and $\Pr(\mathbf{Y}_n|M_j)^{-1}$ for $\tau_k \in \mathcal{T}_0(p_0)$ and $\tau_j \notin \mathcal{T}_0(p_0)$, which converges to 0 in probability by Lemma 1 and (1). As the computational time for $\Pr(\mathbf{Y}_n|M_k)$ grows at the speed of $O(n)$ for $k = 1, \dots, K_n$, that for the refinement stage grows with the sample size at the speed of $O(nK_n)$.

# 3 Simulations

## 3.1 Data sequence without spikes

To evaluate the performance of the proposed BMS method in the settings without spike points, we generate data from two different models. Model I takes the form of

$$\text{Model I}: \ Y_i = \mathbf{h}^\top \mathbf{J}(x_i) + \sigma \epsilon_i,$$

where $\mathbf{h} = (2.01, -2.51, 1.51, -2.01, 2.51, -2.11, 1.05, 2.16, -1.56, 2.56, -2.11)^\top$ with $p_0 = 11$, the error $\epsilon_i \sim N(0, 1)$, and $\sigma = 0.5$. We set $\mathbf{J}(x_i) = \{(1 + \text{sgn}(nx_i - t_j))/2, j = 1, \ldots, p_0\}^\top$, where $\text{sgn}(\cdot)$ is a sign function, and the $x_i$'s are equally spaced on [0, 1]. The true change points are $(t_j/n, j = 1, \ldots, p_0) = (0.1, 0.13, 0.15, 0.23, 0.25, 0.40, 0.44, 0.65, 0.76, 0.78, 0.81)$. The errors are generated from three distributions: $N(0, 1)$; $t$ distribution with 5 degrees of freedom $t(5)$, standardized to have variance 1; and the log-normal distribution $LN(0, 1)$, standardized to have variance 1. Model II considers heteroscedastic errors across segments,

$$\text{Model II}: \ Y_i = \mathbf{h}^\top \mathbf{J}(t_i) + \sigma \epsilon_i \prod_{j=1}^{\mathbf{1}^\top \mathbf{J}(t_i)} v_j,$$

where $(v_j, j = 1, \ldots, 11) = (1, 0.5, 3, 2/3, 0.5, 3, 2/3, 0.5, 3, 2/3, 0.5)$. Other specifications remain the same as those in model I. The over- and under-segmentation errors are respectively defined as

$$d(\widehat{\mathcal{G}}_n | \mathcal{G}_n) = \sup_{b \in \mathcal{G}_n} \inf_{a \in \widehat{\mathcal{G}}_n} |a - b|, \quad d(\mathcal{G}_n | \widehat{\mathcal{G}}_n) = \sup_{b \in \widehat{\mathcal{G}}_n} \inf_{a \in \mathcal{G}_n} |a - b|.$$

For the BMS procedure, we consider three different priors for $\pi(\cdot)$, corresponding to the local prior, non-local moment prior and non-local inverse moment prior. Figure 2 presents the relationship between the maximum of the over- and under-segmentation errors, $|\widehat{p} - p_0|$ and the value of $h$ with sample size 1000, which indicates $h = 0.65$ leading to the smallest segmentation error. We take the minimum distance between candidate points $n_I = \{\log(n)\}^{1.5} h$, where $h \geqslant 0.5$ generally works well in the simulations.

Furthermore, we assess the performance of BMS using different priors under model I with a normal error, when $p_0$ is not prespecified. In Figure 3, we present the selection error which is defined as the maximum of the number of selected change points that are not in $\mathcal{T}_0(p_0)$ and the number of true change points that are not in $\widehat{\mathcal{T}}(\widehat{p})$. The tuning parameters are calibrated to yield the smallest segmentation error and $|\widehat{p} - p_0|$ on average for each prior. Both the selection error and $|\widehat{p} - p_0|$ decrease as the sample size increases, and the prior $\pi_I(\cdot)$ leads to the best convergence among the three prior choices.

For a comprehensive comparison with existing methods, we assess BMS under the non-local inverse moment prior $\pi_I(\cdot)$ with $q = \nu = 2$ and $s = 6$ against existing methods including PELT [12], WBS [7], NOT with normal or heavy-tail distributions [1] and SML [5]. Table 1 summarizes the numerical results under model I and model II with normal, Student's $t$, and log-normal error distributions and their heteroscedastic counterparts. On average, BMS performs the best in selecting the number of change points and balancing both over- and under-segmentation errors. It is expected that the performances of WBS, PELT, and SML deteriorate when the errors do not follow a normal distribution, because they all rely upon parametric model assumptions and thus are not robust to model misspecifications. In contrast, both BMS and NOT behave well under various error distributions. Also, NOT and SML perform the best in controlling the over-segmentation errors, while the resulting estimator $\widehat{p}$ tends to be larger than the true $p_0$. On the other hand, BMS allows for slightly larger over-segmentation errors in order to maintain $\widehat{p}$ to be more concentrated around $p_0$.

## 3.2 Data sequence with spikes

We further evaluate the BMS, NOT and SML methods based on the data sequences contaminated with spike points. Assuming normal noises, we generate 500 sequences and each contains $n = 1000$ points with mean changes of $0.01$ and $-0.01$ at the 400th and 440th observations, respectively. We set the standard deviation of the noise to be 0.002. We further generate 10 random samples uniformly in the ranges of $(-0.07, -0.08)$ and $(0.07, 0.08)$, and add them to the original sequence at random

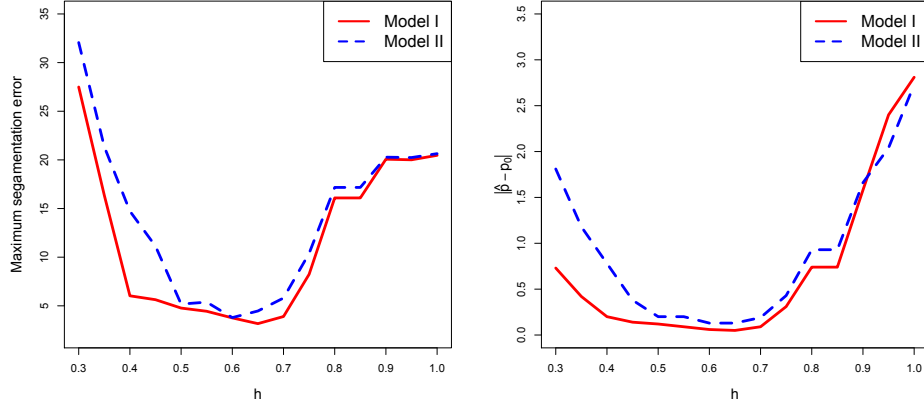

Figure 2: The maximum segmentation error (left) and $|\hat{p} - p_0|$ (right) versus $h$ (the tuning parameter in the minimum distance between candidate points) over 100 simulations with sample size $n = 1000$.

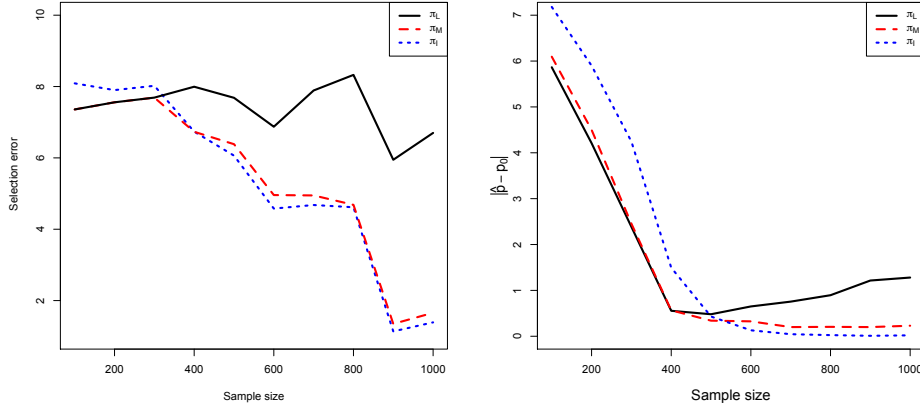

Figure 3: The selection error (left) and $|\hat{p} - p_0|$ (right) averaged over 500 simulations under three different prior distributions: the local prior $\pi_L$, non-local moment prior $\pi_M$, and non-local inverse moment prior $\pi_I$.

locations to form the spike points. These configurations are chosen to mimic the real data setting. We implement BMS, NOT, and SML on the simulated samples, and for BMS we select $n_I = 12$ which is the largest integer that is smaller than $0.65\{\log(n)\}^{1.5}$.

Table 2 shows that BMS, resulting in the smallest $|\hat{p} - p_0|$ on average, is insensitive to the spike points. Figure 4 illustrates the change points detection results for three simulated data sequences. It is observed that NOT ignores both the change points with small signal-noise ratios and the spike signals with small segment lengths, because NOT is more appropriate for settings where the segments are of comparable lengths. On the other hand, SML is sensitive to extreme values, as it is developed to handle frequent and irregular change points. It appears that BMS is the most suitable procedure for this case, because not only does it reinforce the minimal segment length to avoid false identification of spike signals but it also retains the minimal segment length to detect change points with small distance gaps.

Table 1: Comparison results averaged over 200 simulations among the BMS, PELT, WBS, NOT and SML methods under models I and II with three error distributions: $N(0,1)$, $t(5)$, and log-normal $LN(0,1)$, and those with heteroscedastic variances. Standard deviations are given in parentheses.

| Error Distribution | Method | $\leqslant -3$ | $-2$ | $-1$ | $0$ | $1$ | $2$ | $\geqslant 3$ | $d(\mathcal{G}_n|\widehat{\mathcal{G}}_n)$ | $d(\widehat{\mathcal{G}}_n|\mathcal{G}_n)$ |
|---|---|---|---|---|---|---|---|---|---|---|
| | | | | | $\widehat{p}-p_0$ | | | | | |
| $N(0,1)$ | BMS | 0 | 0 | 1 | 197 | 2 | 0 | 0 | 2.41 (6.06) | 1.96 (3.94) |
| | PELT | 0 | 1 | 37 | 162 | 0 | 0 | 0 | 0.91 (1.19) | 6.32 (11.92) |
| | WBS | 0 | 0 | 0 | 194 | 6 | 0 | 0 | 1.22 (4.13) | 0.86 (0.79) |
| | NOT | 0 | 0 | 0 | 192 | 7 | 1 | 0 | 1.93 (8.04) | 0.75 (0.80) |
| | SML | 0 | 0 | 0 | 132 | 52 | 13 | 3 | 12.94 (42.98) | 0.78 (0.90) |
| $t(5)$ | BMS | 0 | 0 | 8 | 190 | 2 | 0 | 0 | 2.15 (5.76) | 2.83 (7.01) |
| | PELT | 0 | 4 | 31 | 165 | 0 | 0 | 0 | 0.95 (1.03) | 6.24 (12.24) |
| | NOT | 0 | 0 | 3 | 184 | 3 | 6 | 4 | 7.57 (27.70) | 1.51 (2.57) |
| | SML | 0 | 0 | 0 | 42 | 34 | 44 | 80 | 40.13 (53.68) | 0.88 (0.87) |
| $LN(0,1)$ | BMS | 0 | 0 | 12 | 180 | 6 | 1 | 2 | 3.69 (12.12) | 3.11 (6.89) |
| | PELT | 1 | 2 | 21 | 135 | 15 | 23 | 3 | 12.10 (29.63) | 7.22 (13.32) |
| | NOT | 0 | 1 | 4 | 183 | 7 | 1 | 4 | 6.06 (26.32) | 1.18 (4.45) |
| | SML | 0 | 0 | 0 | 0 | 0 | 4 | 196 | 111.77 (52.05) | 0.73 (1.33) |
| Hetero-scedastic $N(0,1)$ | BMS | 0 | 0 | 13 | 176 | 8 | 3 | 0 | 3.69 (7.08) | 3.88 (7.36) |
| | PELT | 0 | 0 | 31 | 169 | 0 | 0 | 0 | 1.49 (1.53) | 6.15 (11.44) |
| | NOT | 0 | 0 | 0 | 150 | 23 | 21 | 6 | 7.52 (12.60) | 1.66 (1.68) |
| | SML | 0 | 0 | 0 | 119 | 55 | 20 | 6 | 6.75 (23.98) | 1.37 (1.52) |
| Hetero-scedastic $t(5)$ | BMS | 0 | 0 | 14 | 181 | 4 | 1 | 0 | 2.79 (4.87) | 4.21 (8.17) |
| | PELT | 0 | 4 | 35 | 159 | 2 | 0 | 0 | 1.50 (1.83) | 7.76 (13.48) |
| | NOT | 0 | 1 | 5 | 179 | 10 | 2 | 3 | 8.15 (25.01) | 2.36 (4.70) |
| | SML | 0 | 0 | 0 | 43 | 22 | 41 | 94 | 26.74 (35.42) | 1.27 (1.59) |
| Hetero-scedastic $LN(0,1)$ | BMS | 0 | 0 | 20 | 173 | 7 | 0 | 0 | 3.73 (11.72) | 4.20 (7.85) |
| | PELT | 0 | 2 | 21 | 142 | 22 | 12 | 1 | 6.07 (16.09) | 8.10 (13.93) |
| | NOT | 0 | 0 | 4 | 183 | 7 | 4 | 2 | 6.32 (24.67) | 1.42 (3.70) |
| | SML | 0 | 0 | 0 | 2 | 1 | 0 | 197 | 68.05 (47.15) | 0.87 (1.67) |

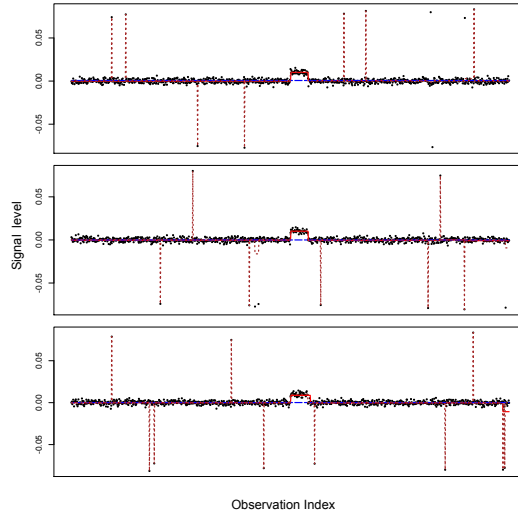

Figure 4: Detection of change points for three simulated data sequences with spike points using BMS (the red solid line), NOT (the blue dashed line) and SML (the brown dotted line).

## 4 MRgRT data

We illustrate the BMS method with application to the MRgRT data which contain 2265 observations ordered by the distances from the sources of the radiations. The R code for implementing the BMS method can be downloaded from our GitHub repository [10]. Throughout the implementation, we use the non-local inverse moment prior $\pi_I(\mu)$ with $q = 2$ and $\nu = 2$, and we set $n_I = 13$.

Table 2: Comparison results averaged over 500 simulations among the BMS, NOT and SML methods based on the data sequences with spike points.

| Method | $\leqslant -3$ | $-2$ | $-1$ | $\widehat{p} - p_0$ 0 | 1 | 2 | $\geqslant 3$ |
|---|---|---|---|---|---|---|---|
| BMS | 0 | 0 | 31 | 276 | 113 | 67 | 13 |
| NOT | 0 | 0 | 387 | 32 | 20 | 11 | 50 |
| SML | 0 | 0 | 0 | 0 | 0 | 0 | 500 |

By varying $s$ from 2 to 10, the left panel in Figure 5 shows that when $s$ is small, BMS identifies more change points, and as $s$ grows the number of identified change points decreases. This phenomenon is consistent with Lemma 4 that the convergence rate for the non-local prior is $O_p\{\exp(-n_I^{s/(1+s)})\}$. When $s$ is small, the Bayes factor vanishes slowly and hence the algorithm picks more spurious change points. When $s$ is sufficiently large, the convergence rate approaches $O_p\{\exp(-n_I)\}$, and hence the algorithm eliminates the flat points more effectively.

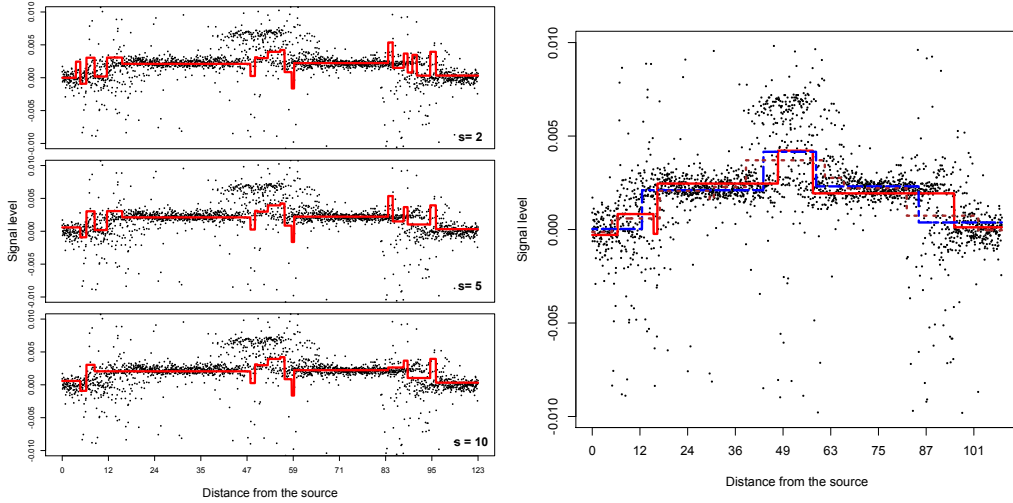

Figure 5: The left panel shows the detection of change points using BMS with $s = 2, 5$, and 10 respectively; the right panel shows the detection of change points after restricting the data in the range of $(-0.01, 0.01)$ using BMS with $s = 10$ (the red solid line), NOT (the blue dashed line), and SML (the brown dotted line).

We further remove the spike points, and thus keep the data within the range of $(-0.01, 0.01)$. By fixing $s = 10$, we implement BMS, NOT and SML on the truncated data sequence. The right panel in Figure 5 shows that the change point detection results using the three methods are largely overlapped. The BMS and NOT methods lead to similar results, and both outperform SML. This implies that removing the spike points improves the accuracy of boundary detection for all the three methods.

## 5 Conclusion

The proposed BMS method can consistently identify multiple mean changes in a data sequence, which effectively removes the flat points without sacrificing the detection accuracy. Our method is particularly useful when the data sequence contains spike points that are not of interest as they are not real change points. The BMS is applied to analyze the MRgRT data for detecting mean changes in the signals, while the NOT, SMT and other methods fail to correctly detect the boundaries. We explore the performance of BMS with different tuning parameters, and the resulting patterns are consistent with the theoretical properties. Moreover, we demonstrate the robustness of BMS to various error distributions.

**Acknowledgment**

The authors would like to thank Dr. Zhou Shouhao from Department of Biostatistics, M.D. Anderson Cancer Center for providing the data. The research is partially supported by grants from the Research Grants Council of Hong Kong (grant number 27304117 for Jiang and 17326316 for Yin).

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
