[Supplementary Material · supp2018.pdf]

# Appendix

## A   Additional figures

Figure A.1: Detection of the dose changes in the DMOS system using the SML method with a normal prior and the maximal number of change points of 30, and the NOT method.

## B   Proofs

Let $g'(\mu)$ and $g''(\mu)$ denote the first and second derivatives of a generic function $g(\mu)$ with respect to $\mu$ respectively, and further define the utility function as

$$U_k(\mu) = -\sum_{l=\tau_k}^{\tau_{k+1}-1}(Y_l - \bar{Y}_{\tau_k} - \mu)^2.$$

The following conditions are imposed for the theoretical derivations.

(A1)  Assume $\mu$ to be in a closed set of points in $\mathbb{R}$.

(A2)  Assume $\pi(\mu)$ to be a continuous density function with bounded first and second derivatives.

(A3)  Assume that $\Pr\{\mathbf{Y}_n | \mathcal{T}(p)\}$ has a unique maximizer in the neighborhood of $\mathcal{T}_0(p_0)$.

**Lemma 1.**  *Assume that $\tau_k$ is a change point for which the mean of $Y_l - \bar{Y}_{\tau_k}$ satisfies $|\mu_{k0}| > \delta$ for $\delta > 0$, $n_k^{1/2}\delta/\sigma \to \infty$, then there is a constant $D > 0$ such that*

$$\lim_{n\to\infty}\Pr\left[\frac{\int\prod_{l=\tau_k}^{\tau_{k+1}-1}\exp\{-(Y_l - \bar{Y}_{\tau_k} - \mu)^2\}\pi(\mu)d\mu}{\prod_{l=\tau_k}^{\tau_{k+1}-1}\exp\{-(Y_l - \bar{Y}_{\tau_k})^2\}} > \exp(Dn_k\delta^2)\right] = 1.$$

Proof: By the definition of $U_k(\mu)$, we can write

$$\frac{\int \prod_{l=\tau_k}^{\tau_{k+1}-1} \exp\{-(Y_l - \bar{Y}_{\tau_k} - \mu)^2\}\pi(\mu)d\mu}{\prod_{l=\tau_k}^{\tau_{k+1}-1} \exp\{-(Y_l - \bar{Y}_{\tau_k})^2\}} = \frac{\int \exp\{U_k(\mu)\}\pi(\mu)d\mu}{\exp\{U_k(0)\}}.$$

We first define $\mathcal{N}_\delta(\mu_{k0}) = \{\mu : |\mu - \mu_{k0}| < \delta\}$ and denote $\mathcal{N}_\delta^c(\mu_{k0})$ as its compliment, and then show

$$\lim_{n\to\infty} \text{Pr}\left[\sup_{\mu\in\mathcal{N}_\delta^c(\mu_{k0})}\{U_k(\mu) - U_k(\mu_{k0})\} < -Dn_k\delta^2\right] = 1.$$

Note that

$$
\begin{aligned}
U_k(\mu) - U_k(\mu_{k0}) &= n_k\left\{(\mu_{k0}^2 - \mu^2) - n_k^{-1}(\mu_{k0} - \mu)\sum_{l=\tau_k}^{\tau_{k+1}-1} 2(Y_l - \bar{Y}_{\tau_k})\right\}\\
&= n_k\{(\mu_{k0}^2 - \mu^2) - 2(\mu_{k0} - \mu)\mu_{k0} + O_p(n_k^{1/2}|\mu_{k0} - \mu|\sigma_k)\}\\
&= n_k\{-(\mu - \mu_{k0})^2\} + O_p(n_k^{1/2}|\mu_{k0} - \mu|\sigma_k)\\
&\leqslant -n_k\delta^2 + O_p(n_k^{1/2}|\mu_{k0} - \mu|\sigma_k)\\
&= -n_k\delta^2/2 - n_k\delta^2/2 + O_p(n_k^{1/2}|\mu_{k0} - \mu|\sigma_k).
\end{aligned}
$$

As $n_k^{1/2}\delta/\sigma \to \infty$, we have $-n_k\delta^2/2 + O_p(n_k^{1/2}|\mu_{k0} - \mu|\sigma_k) < 0$ with probability 1, and thus

$$\lim_{n\to\infty} \text{Pr}\left[\sup_{\mu\in\mathcal{N}_\delta^c(\mu_{k0})}\{U_k(\mu) - U_k(\mu_{k0})\} < -Dn_k\delta^2\right] = 1.$$

When $\tau_k$ is a change point, let $\mu = 0$, because $|\mu_{k0}| > \delta$, we have

$$\lim_{n\to\infty} \text{Pr}\left[\frac{\exp\{U_k(0)\}}{\exp\{U_k(\mu_{k0})\}} < \exp(-Dn_k\delta^2)\right] = 1. \tag{2}$$

By the Laplace approximation,

$$\int \exp\{U_k(\mu)\}\pi(\mu)d\mu = O_p[-U_k''(\tilde{\mu})^{-1/2}\exp\{U_k(\tilde{\mu})\}\pi(\tilde{\mu})], \tag{3}$$

where $\tilde{\mu}$ is the maximizer of $U_k(\mu) + \log\{\pi(\mu)\}$, and $U_k''(\tilde{\mu}) = O_p(n_j)$. Let $\hat{\mu}$ be the maximizer of $U_k(\mu)$, and then

$$
\begin{aligned}
0 &= L_k'(\tilde{\mu}) + \partial\log\pi(\tilde{\mu})/\partial\mu\\
&= L_k''(\mu^*)(\tilde{\mu} - \hat{\mu}) + \partial\log\pi(\tilde{\mu})/\partial\mu,
\end{aligned}
$$

where $\mu^*$ is a point on the line segment between $\tilde{\mu}$ and $\hat{\mu}$. As $\pi(\mu)$ has two bounded derivatives by condition (A2), $L_k''(\mu^*) = O_p(n_j)$, we have $\tilde{\mu} - \hat{\mu} = O_p(n_j^{-1})$. Therefore, (3) can be written as

$$
\begin{aligned}
\int \exp\{U_k(\mu)\}\pi(\mu)d\mu &= O_p[-U_k''(\hat{\mu})^{-1/2}\exp\{U_k(\hat{\mu})\}\pi(\hat{\mu})]\\
&= O_p[-U_k''(\mu_{k0})^{-1/2}\exp\{U_k(\mu_{k0})\}\pi(\mu_{k0})], \tag{4}
\end{aligned}
$$

where the last equality holds because $\hat{\mu}$ is the least squares estimator. This implies

$$\frac{\exp\{U_k(\mu_{k0})\}}{\int \exp\{U_k(\mu)\}\pi(\mu)d\mu} = O_p(n_k^{1/2}),$$

which in conjunction with (2) leads to

$$\lim_{n\to\infty} \text{Pr}\left[\frac{\exp\{U_k(0)\}}{\int \exp\{U_k(\mu)\}\pi(\mu)d\mu} < \exp(-Dn_k\delta^2)\right] = 1.$$

By condition (A2) and the boundedness of $U_k(\mu)$, we have

$$\lim_{n\to\infty} \text{Pr}\left[\frac{\int \exp\{U_k(\mu)\}\pi(\mu)d\mu}{\exp\{U_k(0)\}} > \exp(Dn_k\delta)\right] = 1,$$

which completes the proof.

**Lemma 2.** *Let $\pi(\mu) = \pi_L(\mu)$ be a local prior, and assume that $\tau_j$ is not a change point, i.e., $\mu_{j0} = 0$, then*

$$\frac{\int \prod_{l=\tau_j}^{\tau_{j+1}-1} \exp\{-(Y_l - \bar{Y}_{\tau_j} - \mu)^2\}\pi(\mu)d\mu}{\prod_{l=\tau_j}^{\tau_{j+1}-1} \exp\{-(Y_l - \bar{Y}_{\tau_j})^2\}} = O_p(n_j^{-1/2}).$$

Proof: By the definition of $U_j(\mu)$, we can write

$$\frac{\int \exp\{U_j(\mu)\}\pi(\mu)d\mu}{\exp\{U_j(0)\}} = \frac{\int \prod_{l=\tau_j}^{\tau_{j+1}-1} \exp\{-(Y_l - \bar{Y}_{\tau_j} - \mu)^2\}\pi(\mu)d\mu}{\prod_{l=\tau_j}^{\tau_{j+1}-1} \exp\{-(Y_l - \bar{Y}_{\tau_j})^2\}}.$$

Using the same argument as that leading to (4) with $\mu_{j0} = 0$, we have

$$\int \exp\{U_j(\mu)\}\pi(\mu)d\mu = O_p[-U''(0)^{-1/2} \exp\{U_j(0)\}\pi(0)].$$

As $U''(0)^{-1/2} = O_p(n_j^{-1/2})$, and $\pi(0)$ is a bounded density, we have

$$\frac{\int \exp\{U_j(\mu)\}\pi(\mu)d\mu}{\exp\{U_j(0)\}} = O_p(n_j^{-1/2}).$$

**Lemma 3.** *Let*

$$\pi(\mu) = \pi_M(\mu) \equiv \frac{\mu^{2v}}{C_M}\pi_b(\mu),$$

*where $C_M$ is a normalizing constant, $\pi_b(\mu)$ with $\pi_b(0) > 0$ is the base prior density with $2v$ finite moments, and bounded first two derivatives in the neighborhood around $0$. Assume that $\tau_j$ is not a change point, i.e., $\mu_{j0} = 0$, then*

$$\frac{\int \prod_{l=\tau_j}^{\tau_{j+1}-1} \exp\{-(Y_l - \bar{Y}_{\tau_j} - \mu)^2\}\pi(\mu)d\mu}{\prod_{l=\tau_j}^{\tau_{j+1}-1} \exp\{-(Y_l - \bar{Y}_{\tau_j})^2\}} = O_p(n_j^{-v-1/2}).$$

Proof: We can write

$$\int \prod_{l=\tau_j}^{\tau_{j+1}-1} \exp\{-(Y_l - \bar{Y}_{\tau_j} - \mu)^2\}\pi(\mu)d\mu = \int \exp\{U_j(\mu) + \log\pi(\mu)\}d\mu.$$

Let $h(\mu) = U_j(\mu) + \log\pi(\mu) = 2v\log(\mu) + \log\{\pi_b(\mu)\} + U_j(\mu)$, and let $\tilde{\mu}$ be the maximizer of $h(\mu)$, then we have

$$2v/\tilde{\mu} + \pi_b'(\tilde{\mu})/\pi_b(\tilde{\mu}) + L_j'(\tilde{\mu}) = 0.$$

If we expand $L_j'(\tilde{\mu})$ around $\hat{\mu}$, the least squares estimator for $\mu_{j0}$, the above equality can be rewritten as

$$2v/n + n^{-1}\tilde{\mu}\pi_b'(\tilde{\mu})/\pi_b(\tilde{\mu}) + L_j''(\mu^*)\tilde{\mu}(\tilde{\mu} - \hat{\mu}) = 0,$$

where $\mu^*$ is a point on the line segment between $\hat{\mu}$ and $\tilde{\mu}$. Therefore,

$$\begin{aligned}
O_p(n^{-1}) &= \tilde{\mu}(\tilde{\mu} - \hat{\mu}) \\
&= (\tilde{\mu} - \hat{\mu})^2 + \hat{\mu}(\tilde{\mu} - \hat{\mu}) \\
&= (\tilde{\mu} - \hat{\mu} + \hat{\mu}/2)^2 - \hat{\mu}^2/4.
\end{aligned}$$

Along with the fact that $\hat{\mu} = O_p(n_j^{-1/2})$, we have $\tilde{\mu} - \hat{\mu} = O_p(n_j^{-1/2})$, and $\tilde{\mu} = O_p(n^{-1/2})$. Next, by the Laplace expansion, we have

$$\int \exp\{h(\mu)\}du = O_p(\{2v/\tilde{\mu}^2 - U_j''(\tilde{\mu})\}^{-1/2} \exp[2v\log(\tilde{\mu}) + \log\{\pi_b(\tilde{\mu})\} + U_j(\tilde{\mu})]),$$

and also

$$
\begin{aligned}
n^{-1}U_j(\widetilde{\mu}) - n^{-1}U_j(0) &= n^{-1}U_j'(\mu^\dagger)\widetilde{\mu} \\
&= n^{-1}\{U_j'(\widehat{\mu}) + U_j'(\mu^\dagger) - U_j'(\widehat{\mu})\}\widetilde{\mu} \\
&= O_p(\mu^\dagger - \widehat{\mu})\widetilde{\mu} \\
&= O_p(n^{-1}),
\end{aligned}
\tag{5}
$$

where $\mu^\dagger$ is a point on the line segment between $\widetilde{\mu}$ and $0$. Thus,

$$
|U_j(\widetilde{\mu}) - U_j(0)| = O_p(1).
$$

As a result,

$$
\begin{aligned}
&\frac{\int \prod_{l=\tau_j}^{\tau_{j+1}-1} \exp\{-(Y_l - \bar{Y}_{\tau_j} - \mu)^2\}\pi(\mu)d\mu}{\prod_{l=\tau_j}^{\tau_{j+1}-1} \exp\{-(Y_l - \bar{Y}_{\tau_j})^2\}} \\
&= \frac{\int \exp\{U_j(\mu)\}\pi(\mu)d\mu}{\exp\{U_j(0)\}} \\
&= \frac{\int \exp\{h(u)\}du}{\exp\{U_j(0)\}} \\
&= O_p(\{2v/\widetilde{\mu}^2 - U_j''(\widetilde{\mu})\}^{-1/2} \exp[2v\log(\widetilde{\mu}) + \log\{\pi_M(\widetilde{\mu})\} + U_j(\widetilde{\mu}) - U_j(0)]) \\
&= O_p(n_j^{-1/2}\widetilde{\mu}^{2v}) \\
&= O_p(n_j^{-1/2-v}),
\end{aligned}
$$

where the last equality holds due to the fact that $\widetilde{\mu} = O_p(n^{-1/2})$. This completes the proof.

**Lemma 4.** *Let*

$$
\pi(\mu) = \pi_I(\mu) \equiv \frac{s\nu^{q/2}}{\Gamma\{q/(2s)\}}\mu^{-(q+1)}\exp\left\{-\left(\frac{\mu^2}{\nu}\right)^{-s}\right\}.
$$

*Assume that $\tau_j$ is not a change point, i.e., $\mu_{j0} = 0$, then*

$$
\frac{\int \prod_{l=\tau_j}^{\tau_{j+1}-1} \exp\{-(Y_l - \bar{Y}_{\tau_j} - \mu)^2\}\pi(\mu)d\mu}{\prod_{l=\tau_j}^{\tau_{j+1}-1} \exp\{-(Y_l - \bar{Y}_{\tau_j})^2\}} = O_p\{\exp(-n_j^{s/(s+1)})\}.
$$

Proof: We first write

$$
\int \prod_{l=\tau_j}^{\tau_{j+1}-1} \exp\{-(Y_l - \bar{Y}_{\tau_j} - \mu)^2\}\pi(\mu)d\mu = c\int \exp\left\{U_j(\mu) - \mu^{-2s}\nu^s - (q+1)\log(\mu)\right\}d\mu,
$$

where $c$ is a constant. Let $h(\mu) = U_j(\mu) - \mu^{-2s} - (q+1)\log(\mu)$, and assume $\widetilde{\mu}$ is the maximizer of $h(\mu)$, then we have

$$
U_j'(\widetilde{\mu}) + 2s\widetilde{\mu}^{-2s-1}\nu^s - (q+1)\widetilde{\mu}^{-1} = U_j'(\mu^*)(\widetilde{\mu} - \widehat{\mu}) + 2s\widetilde{\mu}^{-2s-1}\nu^s - (q+1)\widetilde{\mu}^{-1} = 0,
$$

where $\mu^*$ is a point on the line segment between $\widetilde{\mu}$ and $\widehat{\mu}$. The above equality yields

$$
n_j\widetilde{\mu}^{2s+2}(1 - \widehat{\mu}/\widetilde{\mu}) = \frac{2s\nu^s - (q+1)\widetilde{\mu}^{2s}}{-U_j''(\mu^*)/n_j},
\tag{6}
$$

which implies $\widetilde{\mu} = O_p(n_j^{1/(2s+2)})$.

From (6), we have $n\widetilde{\mu}^{2s+1}(\widetilde{\mu} - \widehat{\mu}) = O_p(1)$, which leads to

$$
\widetilde{\mu} - \widehat{\mu} = O_p\{n_j^{-(4s+3)/(2s+2)}\}.
\tag{7}
$$

Following (30) in [11] and using our notation, we obtain

$$\int \exp\{h(\mu)\}du = O_p\left[\left\{\frac{(4s^2+2s)^{2s+2}}{\widetilde{\mu}} - U_j''(\widetilde{\mu})\right\}^{-1/2} |\widetilde{\mu}|^{-q-1}\exp\{-\widetilde{\mu}^{-2s}\nu^s + U_j(\widetilde{\mu})\}\right].$$

Expanding $U_j(\widetilde{\mu})$ around the least squares estimator $\widehat{\mu}$, we have

$$
\begin{aligned}
U_j(\widetilde{\mu}) &= U_j(\widehat{\mu}) + 1/2 U_j''(\mu^*)(\widetilde{\mu}-\widehat{\mu})^2 \\
&= U_j(\widehat{\mu}) + o_p(1) \\
&= U_j(0) + O_p(1),
\end{aligned}
$$

where $\mu^*$ is a point on the line segment between $\widetilde{\mu}$ and $\widehat{\mu}$, the second equality follows (7), and the last equality follows the same argument as that leading to (5). Therefore, we have

$$
\begin{aligned}
&\frac{\int \prod_{l=\tau_j}^{\tau_{j+1}-1}\exp\{-(Y_l-\bar{Y}_{\tau_j}-\mu)^2\}\pi(\mu)d\mu}{\prod_{l=\tau_j}^{\tau_{j+1}-1}\exp\{-(Y_l-\bar{Y}_{\tau_j})^2\}} \\
&= \frac{\int \exp\{U_j(\mu)\}\pi(\mu)d\mu}{\exp\{U_j(0)\}} \\
&= \frac{\int \exp\{h(u)\}du}{\exp\{U_j(0)\}} \\
&= O_p\left[\left\{\frac{(4s^2+2s)^{2s+2}}{\widetilde{\mu}} - U_j''(\widetilde{\mu})\right\}^{-1/2} |\widetilde{\mu}|^{-q-1}\exp\{-\widetilde{\mu}^{-2s}\nu^s + U_j(\widetilde{\mu}) - U_j(0)\}\right] \\
&= O_p\{\exp(-n_j^{s/(s+1)})\},
\end{aligned}
$$

which completes the proof.

**Lemma 5.** *Assume $p_0=1$ and $\tau_k$ is the only true change point. As $n_k^{1/2}\delta/\sigma \to \infty$, $\Pr(M_k|\mathbf{Y}_n)-1 = O_p\{K_n a_{n_I}\exp(-n_I\delta^2)\}$. Hence when $n_I/\log(n)\to c>0$, $n_I \leqslant \lambda$, we have $\Pr(M_k|\mathbf{Y}_n) \xrightarrow{p} 1$.*

Proof: First, we can write

$$\Pr(M_k|\mathbf{Y}_n) = \left\{1 + \sum_{j\neq k}^{K_n}\frac{\Pr(\mathbf{Y}_n|M_j)}{\Pr(\mathbf{Y}_n|M_k)}\right\}^{-1}. \tag{8}$$

To show $\Pr(M_k|\mathbf{Y}_n) - 1 \to 0$, it is equivalent to showing

$$\sum_{j=1,j\neq k}^{K_n}\frac{\Pr(\mathbf{Y}_n|M_j)}{\Pr(\mathbf{Y}_n|M_p)} \to 0.$$

Note that

$$\frac{\Pr(\mathbf{Y}_n|M_j)}{\Pr(\mathbf{Y}_n|M_k)} = A \times B,$$

where

$$A = \frac{\int \prod_{l=\tau_j}^{\tau_{j+1}-1}\exp\{-(Y_l-\bar{Y}_{\tau_j}-\mu)^2\}\pi(\mu)d\mu}{\prod_{l=\tau_j}^{\tau_{j+1}-1}\exp\{-(Y_l-\bar{Y}_{\tau_j})^2\}}$$

and

$$B = \frac{\prod_{l=\tau_k}^{\tau_{k+1}-1}\exp\{-(Y_l-\bar{Y}_{\tau_k})^2\}}{\int \prod_{l=\tau_k}^{\tau_{k+1}-1}\exp\{-(Y_l-\bar{Y}_{\tau_k}-\mu)^2\}\pi(\mu)d\mu}.$$

As shown in [11], $A$ is a Bayes factor whose convergence rate is $O_p(a_{n_j})$. For $B$, note that the data in $[\tau_k,\tau_{k+1})$ are generated from the model with mean $\mu_{k0}$ such that $|\mu_{k0}|>\delta$. Hence, we have

$$B = O_p\{\exp(-n_k\delta^2)\},$$

where the last equality holds by Lemma 1 that

$$\lim_{n\to\infty} \left( \mathrm{Pr}\left[ \frac{\prod_{l=\tau_k}^{\tau_{k+1}-1} \exp\{-(Y_l - \bar{Y}_{\tau_k})^2\}}{\int \prod_{l=\tau_k}^{\tau_{k+1}-1} \exp\{-(Y_l - \bar{Y}_{\tau_k} - \mu)^2\}\pi(\mu)d\mu} < \exp(-Dn_k\delta^2) \right] \right) = 1,$$

where $D$ is a constant. Combining the convergence rates for $A$ and $B$, we have

$$AB = O_p\{a_{n_k}\exp(-n_k\delta^2)\}.$$

Thus, this leads to

$$\frac{\mathrm{Pr}(\mathbf{Y}_n|M_j)}{\mathrm{Pr}(\mathbf{Y}_n|M_k)} = AB = O_p\{a_{n_I}\exp(-n_I\delta^2)\},$$

and

$$\sum_{j=1}^{K_n} \frac{\mathrm{Pr}(\mathbf{Y}_n|M_j)}{\mathrm{Pr}(\mathbf{Y}_n|M_k)} - 1 = O_p\{K_n a_{n_I}\exp(-n_I\delta^2)\}$$

Plugging this result into (8), we have

$$\mathrm{Pr}(M_k|\mathbf{Y}_n) \xrightarrow{p} 1,$$

which completes the proof.

**Proof of Theorem 1**
First, we can write

$$\sum_{M_k\in\mathcal{M}} \mathrm{Pr}(M_k|\mathbf{Y}_n) = \left\{ 1 + \frac{\sum_{M_j\notin\mathcal{M}} \mathrm{Pr}(\mathbf{Y}_n|M_j)}{\sum_{M_k\in\mathcal{M}} \mathrm{Pr}(\mathbf{Y}_n|M_k)} \right\}^{-1}.$$

Note that

$$\frac{\sum_{M_j\notin\mathcal{M}} \mathrm{Pr}(\mathbf{Y}_n|M_j)}{\sum_{M_k\in\mathcal{M}} \mathrm{Pr}(\mathbf{Y}_n|M_k)} \leqslant \frac{\sum_{M_j\notin\mathcal{M}} \mathrm{Pr}(\mathbf{Y}_n|M_j)}{\mathrm{Pr}(\mathbf{Y}_n|M_k)},$$

for $M_k \in \mathcal{M}$. Hence, by the same argument as that leading to Lemma 5, we have

$$\frac{\sum_{M_j\notin\mathcal{M}} \mathrm{Pr}(\mathbf{Y}_n|M_j)}{\sum_{M_k\in\mathcal{M}} \mathrm{Pr}(\mathbf{Y}_n|M_k)} = O_p\{K_n a_{n_I}\exp(-n_I\delta^2)\},$$

and

$$\sum_{M_k\in\mathcal{M}} \mathrm{Pr}(M_k|\mathbf{Y}_n) - 1 = O_p\{K_n a_{n_I}\exp(-n_I\delta^2)\},$$

which completes the proof.

**Proof of Proposition 1**
Following [15], we define $x$ as an $n_I$-flat point so that there is no change-point in $(x - n_I, x + n_I)$.
Let $\mathcal{F}$ be the set of all $n_I$-flat points, then

$$\mathrm{Pr}\left\{ \left( \bigcap_{t\in\mathcal{T}_0(p_0)} R_t > C \right) \bigcap \left( \bigcap_{\tau\in\mathcal{F}} R_\tau < C \right) \right\}$$

$$= 1 - \mathrm{Pr}\left\{ \left( \bigcup_{\tau\in\mathcal{T}_0(p_0)} R_\tau > C \right) \bigcup \left( \bigcup_{\tau\in\mathcal{F}} R_\tau < C \right) \right\}$$

$$= 1 - \mathrm{Pr}\left\{ \left( \max_{t\in\mathcal{T}_0(p_0)} R_t > C \right) \bigcup \left( \min_{\tau\in\mathcal{F}} R_\tau < C \right) \right\}$$

$$\geqslant 1 - \left\{ \mathrm{Pr}\left( \max_{t\in\mathcal{T}_0(p_0)} R_t > C \right) + \mathrm{Pr}\left( \min_{\tau\in\mathcal{F}} R_\tau < C \right) \right\}.$$

For each $\tau \in \mathcal{T}_0(p_0)$,

$$\Pr(R_\tau < C) = O\{\exp(-n_I \delta^2)\},$$

by Lemma 1. Furthermore, for $\tau \in \mathcal{F}$,

$$\Pr(R_\tau > C) = O(a_{n_I}),$$

by Lemmas 2–4. Hence,

$$\Pr\left\{\left(\bigcap_{t \in \mathcal{T}_0(p_0)} R_t > C\right)\bigcap\left(\bigcap_{\tau \in \mathcal{F}} R_\tau < C\right)\right\} \geqslant 1 - O[\min\{\exp(-n_I\delta^2), a_{n_I}\}].$$

By Lemma 3 in [15], for any $t \in \mathcal{T}_0(p_0)$ we have a $\tau \in \mathcal{H}_c(n_I)$ such that

$$\Pr\{t \in (\tau - n_I, \tau + n_I)\} = 1 - O[\min\{\exp(-n_I\delta^2), a_{n_I}\}].$$

**Proof of Theorem 2**
We first show that for a given $p$, $\widehat{\mathcal{T}}(p)$ is the maximizer of $\Pr\{\mathbf{Y}_n | \mathcal{T}(p)\}$. Based on the BMS procedure, $\widehat{\mathcal{T}}(p)$ is the maximizer of $\sum_{M_k \in \mathcal{M}} \Pr(M_k | \mathbf{Y}_n)$, where $\mathcal{M} = \{M_k, \tau_k \in \mathcal{T}(p)\}$. As we impose the uniform prior on $M_k$, $\widehat{\mathcal{T}}(p)$ is the maximizer of

$$\sum_{M_k \in \mathcal{M}} \Pr(\mathbf{Y}_n | M_k) \tag{9}$$

$$= D_n \sum_{M_k \in \mathcal{M}} \prod_{j=1, j\neq k}^{K} \prod_{l=\tau_j}^{\tau_{j+1}-1} \exp\{-(Y_l - \bar{Y}_{\tau_j})^2\} \int \prod_{l=\tau_k}^{\tau_{k+1}-1} \exp\{-(Y_l - \bar{Y}_{\tau_k} - \mu)^2\}\pi(\mu)d\mu$$

$$= D_n \prod_{j=1}^{K} \prod_{l=\tau_j}^{\tau_{j+1}-1} \exp\{-(Y_l - \bar{Y}_{\tau_j})^2\} \sum_{M_k \in \mathcal{M}} \frac{\int \prod_{l=\tau_k}^{\tau_{k+1}-1} \exp\{-(Y_l - \bar{Y}_{\tau_k} - \mu)^2\}\pi(\mu)d\mu}{\prod_{l=\tau_k}^{\tau_{k+1}-1} \exp\{-(Y_l - \bar{Y}_{\tau_k})^2\}}$$

where $D_n$ is a constant depending on $n$. Further note that

$$\Pr\{\mathbf{Y}_n | \mathcal{T}(p)\} \tag{10}$$

$$= \prod_{\tau_j \notin \mathcal{T}(p)} \prod_{l=\tau_j}^{\tau_{j+1}-1} \exp\{-(Y_l - \bar{Y}_{\tau_j})^2\} \prod_{\tau_k \in \mathcal{T}(p)} \int \prod_{l=\tau_k}^{\tau_{k+1}-1} \exp\{-(Y_l - \bar{Y}_{\tau_k} - \mu)^2\}\pi(\mu)d\mu$$

$$= \prod_{j=1}^{K} \prod_{l=\tau_j}^{\tau_{j+1}-1} \exp\{-(Y_l - \bar{Y}_{\tau_j})^2\} \prod_{\tau_k \in \mathcal{T}(p)} \frac{\int \prod_{l=\tau_k}^{\tau_{k+1}-1} \exp\{-(Y_l - \bar{Y}_{\tau_k} - \mu)^2\}\pi(\mu)d\mu}{\prod_{l=\tau_k}^{\tau_{k+1}-1} \exp\{-(Y_l - \bar{Y}_{\tau_k})^2\}}.$$

Comparing (9) and (10), clearly they have the same optimizer, and thus $\widehat{\mathcal{T}}(p)$ is also the maximizer of (10). Hence, our BMS procedure results in the estimators $\widehat{p}$ and $\widehat{\mathcal{T}}(\widehat{p})$ that maximize $\Pr\{\mathbf{Y}_n | \mathcal{T}(p)\}$.

Next, let $\mathcal{E}_1$ be the event that at least one $j$ such that $t_j \in (\tau_k, \tau_{k+1})$, and $\widehat{t}_i \neq \tau_k$, $\widehat{t}_i \neq \tau_{k+1}$ for all $i$ and $\tau_k, \tau_{k+1} \in \mathcal{H}_c(n_I)$, $\widehat{t}_i \in \widehat{\mathcal{T}}(\widehat{p})$ that maximizes $\Pr\{\mathbf{Y}_n | \mathcal{T}(p)\}$. Following similar arguments as those in [5], we show that the probability of $\mathcal{E}_1$ goes to 0. Suppose that $\widehat{\mathcal{T}}(\widehat{p})$ is such an estimate. Consider the first case where $(t_j - \tau_k + 1)(\tau_{k+1} - \tau_k + 1)^{-1} = O(1)$; that is, $t_j$ is bounded away from $\tau_k$. We can choose a set of change points,

$$\widetilde{\mathcal{T}}(\widehat{p}+1) \equiv \{\widetilde{\tau}_1, \ldots, \widetilde{\tau}_{\widehat{p}+1}\}$$
$$= \{\widehat{t}_1, \ldots, \widehat{t}_i, \tau_{k+1}, \widehat{t}_{i+1}, \widehat{t}_{\widehat{p}}\}.$$

Then,

$$\frac{\Pr\{\mathbf{Y}_n | \widehat{\mathcal{T}}(\widehat{p})\}}{\Pr\{\mathbf{Y} | \widetilde{\mathcal{T}}(\widehat{p}+1)\}} = \frac{\prod_{l=\tau_{k+1}}^{\tau_{k+2}-1} \exp\{-(Y_l - \bar{Y}_{\tau_{k+1}})^2\}}{\int \prod_{l=\tau_{k+1}}^{\tau_{k+2}-1} \exp\{-(Y_l - \bar{Y}_{\tau_{k+1}} - \mu)^2\}\pi(\mu)d\mu}.$$

Because $\lim_{n\to\infty}\sup n_I/\lambda < 1/2$, there is an $N$, such that for all $n > N$, $t_{j+1} > \tau_{k+2}$, and hence there is no change point within $(\tau_{k+1}, \tau_{k+2})$. This prevents the situation where there are more than one change points in between $\tau_k$ and $\tau_{k+2}$. Further for $n > N$,

$$
\begin{aligned}
E(Y_l - \bar{Y}_{\tau_{k+1}}) &= (\tau_{k+1} - \tau_k + 1)^{-1} E\left\{ \sum_{s=\tau_k}^{t_j} (Y_l - Y_s) + \sum_{s=t_j+1}^{\tau_{k+1}} (Y_l - Y_s) \right\} \\
&\geqslant (t_j - \tau_k + 1)(\tau_{k+1} - \tau_k + 1)^{-1}\delta.
\end{aligned}
$$

Therefore, by Lemma 1,

$$
\frac{\Pr\{\mathbf{Y}_n|\widehat{\mathcal{T}}(\widehat{p})\}}{\Pr\{\mathbf{Y}|\widetilde{\mathcal{T}}(\widehat{p}+1)\}} = O_p\{\exp(-n_I\delta^2)\}.
$$

If $(t_j - \tau_k + 1)(\tau_{k+1} - \tau_k + 1)^{-1} = o(1)$, we define

$$
\begin{aligned}
\widetilde{\mathcal{T}}(\widehat{p}+1) &\equiv \{\widetilde{\tau}_1, \ldots, \widetilde{\tau}_{\widehat{p}+1}\} \\
&= \{\widehat{t}_1, \ldots, \widehat{t}_i, \tau_k, \widehat{t}_{i+1}, \widehat{t}_{\widehat{p}}\},
\end{aligned}
$$

and then

$$
\begin{aligned}
\frac{\Pr\{\mathbf{Y}_n|\widehat{\mathcal{T}}(\widehat{p})\}}{\Pr\{\mathbf{Y}|\widetilde{\mathcal{T}}(\widehat{p}+1)\}} &= \frac{\prod_{l=\tau_k}^{\tau_{k+1}-1} \exp\{-(Y_l - \bar{Y}_{\tau_k})^2\}}{\int \prod_{l=\tau_k}^{\tau_{k+1}-1} \exp\{-(Y_l - \bar{Y}_{\tau_k} - \mu)^2\}\pi(\mu)d\mu} \\
&= \frac{\prod_{l=t_j}^{\tau_{k+1}-1} \exp\{-(Y_l - \bar{Y}_{\tau_k})^2\}}{\int \prod_{l=t_j}^{\tau_{k+1}-1} \exp\{-(Y_l - \bar{Y}_{\tau_k} - \mu)^2\}\pi(\mu)d\mu} \\
&\quad \times \frac{\int \prod_{l=t_j}^{\tau_{k+1}-1} \exp\{-(Y_l - \bar{Y}_{\tau_k} - \mu)^2\}\pi(\mu)d\mu}{\int \prod_{l=\tau_k}^{\tau_{k+1}-1} \exp\{-(Y_l - \bar{Y}_{\tau_k} - \mu)^2\}\pi(\mu)d\mu} \\
&\quad \times \frac{\prod_{l=\tau_k}^{\tau_{k+1}-1} \exp\{-(Y_l - \bar{Y}_{\tau_k})^2\}}{\prod_{l=t_j}^{\tau_{k+1}-1} \exp\{-(Y_l - \bar{Y}_{\tau_k})^2\}},
\end{aligned}
$$

where the first term is of order $O_p\{\exp(-n_I\delta^2)\}$ by Lemma 1, and the last two terms are of order $O_p(1)$ because $(t_j - \tau_k + 1)(\tau_{k+1} - \tau_k + 1)^{-1} = o(1)$. Therefore,

$$
\Pr\left[\frac{\Pr\{\mathbf{Y}_n|\widehat{\mathcal{T}}(\widehat{p})\}}{\Pr\{\mathbf{Y}|\widetilde{\mathcal{T}}(\widehat{p}+1)\}} > 1 \Big| \mathcal{E}_1\right] \leqslant E\left[\frac{\Pr\{\mathbf{Y}_n|\widehat{\mathcal{T}}(\widehat{p})\}}{\Pr\{\mathbf{Y}|\widetilde{\mathcal{T}}(\widehat{p}+1)\}}\right] = O\{\exp(-n_I\delta^2)\}.
$$

As $\widehat{\mathcal{T}}(\widehat{p})$ is the maximizer of $\Pr\{\mathbf{Y}_n|\mathcal{T}(p)\}$, we have

$$
\Pr\left[\frac{\Pr\{\mathbf{Y}_n|\widehat{\mathcal{T}}(\widehat{p})\}}{\Pr\{\mathbf{Y}|\widetilde{\mathcal{T}}(\widehat{p}+1)\}} > 1\right] = 1,
$$

because $\widehat{\mathcal{T}}(\widehat{p})$ is the maximizer of $\Pr\{\mathbf{Y}_n|\widehat{\mathcal{T}}(\widehat{p})\}$ and it is unique by condition (A3). By the Bayes rule, we have

$$
\Pr\left[\mathcal{E}_1 \Big| \frac{\Pr\{\mathbf{Y}_n|\widehat{\mathcal{T}}(\widehat{p})\}}{\Pr\{\mathbf{Y}|\widetilde{\mathcal{T}}(\widehat{p}+1)\}} > 1\right] \leqslant O\{\exp(-n_I\delta^2)\}. \tag{11}
$$

Hence, $\Pr(\widehat{t}_i = \tau_k \text{ or } \widehat{t}_i = \tau_{k+1}) = 1 - O\{\exp(-n_I\delta^2)\}$. Further note that $\tau_k$ and $\tau_{k+1}$ are in the $n_I$-neighborhood of $t_j$, and thus for any $t_j$, there is a $\widehat{t}_i$ such that

$$
\Pr\{t_j \in (\widehat{t}_i - n_I, \widehat{t}_i + n_I)\} = 1 - O\{\exp(-n_I\delta^2)\}.
$$

As it holds for any $j$, we can write

$$
\Pr\left\{\sup_{\widehat{t}_j\in\widehat{\mathcal{T}}(\widehat{p})} \inf_{t_j\in\mathcal{T}_0(p_0)} |(\widehat{t}_j - t_j)/n| \leqslant n_I/n\right\} = 1 - O\{\exp(-n_I\delta^2)\}.
$$

Next we show that for any $\hat{t}_i$ there is a $t_j$ in the $n_I$-neighborhood of $\hat{t}_i$. Define $\mathcal{E}_2$ as the event that there is at least one $\hat{t}_i$ such that there is no $t_j$ in the $n_I$-neighborhood of $\hat{t}_i$. Let $\widehat{\mathcal{T}}(\hat{p})$ be such an estimate that $\hat{t}_i$ is the $k$th candidate point, and $(\hat{t}_i, \tau_{k+1})$ and $(\tau_{k-1}, \hat{t}_i)$ do not contain $t_j$ for all $j$. Then, we define a new set of change points by deleting $\hat{t}_i$,

$$\widetilde{\mathcal{T}}(\hat{p}-1) = \{\hat{t}_1, \ldots, \hat{t}_{i-1}, \hat{t}_{i+1}, \hat{t}_{\hat{p}}\}.$$

Then,

$$\frac{\mathrm{Pr}\{\mathbf{Y}_n | \widehat{\mathcal{T}}(\hat{p})\}}{\mathrm{Pr}\{\mathbf{Y} | \widetilde{\mathcal{T}}(\hat{p}-1)\}} = \frac{\prod_{l=\hat{t}_i}^{\tau_{k+1}-1} \exp\{-(Y_l - \bar{Y}_{\hat{t}_i} - \mu)^2\} \pi(\mu) d\mu}{\prod_{l=\hat{t}_i}^{\tau_{k+1}-1} \exp\{-(Y_l - \bar{Y}_{\hat{t}_i})^2\}} = O_p(a_{n_I})$$

by Lemmas 2–4. Therefore, using the same argument as that leading to (11), we have

$$\mathrm{Pr}\left[\mathcal{E}_2 \left| \frac{\mathrm{Pr}\{\mathbf{Y}_n | \widehat{\mathcal{T}}(\hat{p})\}}{\mathrm{Pr}\{\mathbf{Y} | \widetilde{\mathcal{T}}(\hat{p}-1)\}} > 1\right.\right] = O(a_{n_I}).$$

For any $\hat{t}_i$, there exists a $t_j$ such that

$$\mathrm{Pr}\{\hat{t}_i \in (t_j - n_I, t_j + n_I)\} = 1 - O(a_{n_I}).$$

It holds for any $\hat{t}_i$, and thus we have

$$\mathrm{Pr}\left[\sup_{t_j \in \mathcal{T}_0(p_0)} \inf_{\hat{t}_j \in \widehat{\mathcal{T}}(\hat{p})} |(\hat{t}_j - t_j)/n| < n_I/n\right] = 1 - O(a_{n_I}).$$

Because $(t_j - n_I, t_j + n_I)$ contains only one estimate by the definition of $\widehat{\mathcal{T}}(\hat{p})$ that $|\hat{t}_{j+1} - \hat{t}_j| > \lambda$, we have $\mathrm{Pr}(\hat{p} = p_0) = 1 - O_p(\max\{\exp(-n_I \delta^2), a_{n_I}\})$ by using the same arguments as those leading to Theorem 3.3 in [5]. $\qquad\square$