[Reviews · NeurIPS 2018]

Reviewer 1



This paper illustrates how to use the non-local prior to detect change points in the Bayesian framework. The simulation study demonstrates the Bayesian model selection (BMS) is robust to the error distributions. The non-local prior has been used in the Bayesian community and the theoretical foundation is well understood. The paper is easy to read except the abbreviations in Section 3. • In Section 3, the results are discussed assuming that readers know all abbreviations and what they are. It would be nice to spell out the exact name and a brief reason of the behaviour of those methods. • In general, I feel this paper sits on the boundary of statistical journals and NIP. With my limited experience, I question whether the usage of non-local prior and the case study (MRgRT data) are appealing to types of problems that the NIP community interests.

Reviewer 2



This paper proposes a two stage Bayesian model selection procedure to (1) screen for a set of candidate change points, and (2) select a set of change points from that candidate set. The paper gives selection consistency bounds under and over segmentation errors. Then the paper applies the procedure to a set of synthetic data with varying error distributions, and a set of actual magnetic resonance imaging guided radiation therapy data (which has sufficient noisy to break most existing change point detection methods). The paper is generally well written, but some of the details, such as the behavior of local and moment priors, is not explained in the main paper and it is missing from the supplementary material. However, the main strengths of this paper are (1) a new change point detection method that solves some of the noise sensitivity present in other change point methods, (2) strong theoretical results to support that method, and (3) application to a real problem. Image-guided radiation therapy data is usually thoroughly messy and fast, stable inference to detect tissue and tumor regions in a patient that may be moving (including heartbeats, breathing, etc) is a tough problem.

Reviewer 3



===Update:======= I downgrade my review to 5. The main concern is 1) Some more extensive simulations will make the results more convincing, as the numerical experiment is the only way to assess the performance of the proposed priors. 2) I would recommend more theoretical clarifications on the usage of Bayes factor, which from my understanding will only be selection-consistent under M-closed assumption. It might take a major revision to reflect such comprehensive comparisons. With that being said, I believe the paper does contain interesting results that are novel and useful to the community. In particular, the theoretical results seem sound, and the paper is fairly readable. But I think there is also room for improvement. ===Original============ This paper applies Bayesian model selection in the context of boundary detection. Non-local prior is proved to have a higher order convergence rate of Bayes factor BF(1|0) in hypothesis-testing when the null is true. As a consequence, the proposed algorithm will enjoy a quicker rate towards consistency in boundary detection. Theorem 2 mathematically proves the consistency. The experiments show advantages over all other existing methods. The structure of the paper is clear and easy to follow. However, I am not extremely convinced why Bayesian model selection is the most valid approach in the context of change point detection. Particularly in the case of multiple change points (the non-trivial situation), the proposed method is more like Bayesian model averaging (BMA), rather than model selection in the sense that multiple models are selected and combined in the end. Why is the marginal evidence a good indicator for the joint? (Sure, I understand it works in empirical experiments) Furthermore, I am wondering if it would be worth comparing the proposed methods with other model selection methods other than marginal likelihood (for example, leave-one-out log predictive density, which is not sensitive to the construction of priors, though you probably have to modify LOO a little bit for time series). Or equivalently, others model averaging methods (Bayesian leave-one-out stacking, for instance, has shown advantage over BMA in M-open situation).